# Integrating ecosystem benefits for sustainable water allocation in hydroeconomic modeling

Daniel Crespo[1,2], Jose Albiac[2,3,4]*, Ariel Dinar[5], Encarna Esteban[2,6], Taher Kahil[4]

**1** Agrifood Research and Technology Centre (CITA) and University of Zaragoza, Zaragoza, Spain, **2** Instituto Agroalimentario de Aragón (IA2), Zaragoza, Spain, **3** Economics and Business School, University of Zaragoza, Zaragoza, Spain, **4** Water Security Research Group, Biodiversity and Natural Resources Program, International Institute for Applied System Analysis (IIASA), Laxenburg, Austria, **5** School of Public Policy, University of California, Riverside, California, United States of America, **6** School of Humanities and Social Sciences, University of Zaragoza, Teruel, Spain

☯ These authors contributed equally to this work.
* maella@unizar.es

**Data Availability Statement:** All relevant data are within the paper and its Supporting Information files.

**Funding:** J.A received the funding from projects INIA RTA2014-00050-00-00 and INIA RTA2017-

## Abstract

The increasing concern about the degradation of water-dependent ecosystems calls for considering ecosystems benefits in water management decision-making. Sustainable water management requires adequate economic and biophysical information on water systems supporting both human activities and natural ecosystems. This information is essential for assessing the impact on social welfare of water allocation options. This paper evaluates various alternative water management policies by including the spatial and sectoral interrelationships between the economic and environmental uses of water. A hydroeconomic model is developed to analyze water management policies for adaptation to reduced water availability in the Ebro Basin of Spain. The originality in our contribution is the integration of environmental benefits across the basin, by using endemic biophysical information that relates stream flows and ecosystem status in the Ebro Basin. The results show the enhancement of social welfare that can be achieved by protecting environmental flows, and the tradeoffs between economic and environmental benefits under alternative adaptation strategies. The introduction of water markets is a policy that maximizes the private benefits of economic activities, but disregards environmental benefits. The results show that the current institutional policy where stakeholders cooperate inside the water authority, provides lower private benefits but higher environmental benefits compared to those obtained under water markets, especially under severe droughts. However, the water authority is not allocating enough environmental flows to optimize social welfare. This study informs strategies for protection of environmental flows in the Ebro Basin, which is a compelling decision under the imminent climate change impacts on water availability in coming decades.

00082-00-00 of the Ministry for Science and Innovation, partly financed by European ERDF funds (https://www.ciencia.gob.es/site-web/). The funders had no role in study design, data collection and analysis, decision to publish, or preparation of the manuscript.

## Introduction

Ongoing water management and policies across the world deal with water scarcity by reallocating water to the most financially profitable activities or to priority uses (e.g., drinking water), with little or no consideration for the effects of reallocations on aquatic ecosystems [1]. This inadequate recognition of the environmental services and related benefits in water allocation decisions has resulted in the degradation of many valuable ecosystems globally. In fact, biodiversity in inland aquatic ecosystems is the most threatened of all ecosystems [2,3]. Overuse of water resources and ecosystem status deterioration have led to a decline in the environmental services provided by ecosystems [4].

The growing concern about the environment fosters new methodologies for assessing environmental impacts of water degradation and improving policy decision making. Sustainable water management should account for the environmental externalities resulting from water allocation decisions, although the complex response of ecosystems to changes in environmental flows and imprecise environmental valuation make it difficult to identify optimal environmental flow requirements [5].

The expanding withdrawals by economic activities and the declining water availability as a result of droughts and the impending climate change are worsening water scarcity problems in arid and semi-arid regions [6]. Some striking examples are the disappearance of the Aral Sea, the fourth largest inland lake in the world, the desiccation of the Zayandeh Rud river in Isfahan (Iran), and the decades long deterioration of the Colorado river delta in Mexico. Environmental flows must remain at sufficient levels to protect ecosystem health, although this will reduce the water available for economic sectors. The sustainable management of water resources needs scientific knowledge and appropriate governance for enhancing the balance between human water withdrawals and environmental flows in basins [7,8]. Understanding the interactions between humans and rivers is essential for the assessment and implementation of sustainable environmental flows [9].

Freshwater ecosystems provide goods and services to society, which have characteristics of public good or common pool resources. Sustainable water allocation requires identifying and valuing the benefits of environmental services along with the private benefits of economic activities [10]. Current water policies mostly ignore the public good and common pool aspects of environmental flows. Aquifers are examples of common pool resources being overused, with massive global groundwater depletion (300 km$^3$ over 900 km$^3$ of extractions per year [11]), triggering very large ecosystem damages from the degradation of wetlands and the decline of stream flows in basins. Consequently, a more sustainable management must incorporate the external effects of human water withdrawals.

Hydroeconomic modeling is a valuable tool for identifying improved basin-level water management options, especially for adaptation to the impending climate change. This tool integrates several aspects such as the spatial distribution of water resources and the storage and transport infrastructures, water-based economic activities by sector and location, and water-dependent ecosystems. Selected notable works supported by hydroeconomic analysis can be found in Ward [12]. The advantage of hydroeconomic modeling is the linkage between hydrology, economy, environment, and institutions in evaluating water allocation alternatives. The hydrology component represents the supply nodes of both surface and groundwater, and the demand nodes for irrigation, urban and industrial provision, and hydropower production.

Ecosystem protection in hydroeconomic models is usually represented by minimum environmental flows, because of the complexity of modeling the ecological response to streamflows. Some hydroeconomic models analyze environmental and also salinity damages in terms of water savings, replacement costs or crop production damages [13–17]. Recreation benefits

such as boating and fishing are sometimes included in relation to streamflow levels, and travel cost or contingent valuation techniques are used for valuation of the ecosystem services [18–24].

Estimating environmental benefits or damages in riverine areas is difficult due to lack of information, so proxies are used, such as environmental drought cost (measured as an increasing and convex function of drought length) by Grafton et al. [25]. A better option is to analyze the dependence between wet area and streamflow, and then select values for environmental services per unit of wet area from valuation studies [26–29]. However these environmental benefit functions are not based on biophysical processes since the response of ecosystems to stream flows across river reaches is mostly unknown. Very few hydroeconomic studies specify ecological responses based on biophysical principles. Some examples are Yang and Cai [30] that include fish diversity in a multiobjective optimization problem using the Shannon index, and Bryan et al. [31] that undertake a more extensive approach by considering the aggregated response of birds, vegetation and fish based on the biophysical information of inundation dynamics in floodplains.

The benefits of ecosystem services can be estimated by finding the response of ecosystems to water flows, and then valuating the services provided by these ecosystems. Information on environmental benefits in hydroeconomic models is quite limited because of the difficulties in identifying ecosystems and their services, and how these respond to changes in stream flows. Several studies include ecosystem water consumption in hydroeconomic modeling [32,33], however the unspecified response of ecosystems to environmental flows and the scarcity of valuation estimates undermine the accuracy of results.

In any case there is a research gap, because the representation of environmental benefits in hydroeconomic modeling is patchy and limited. The reason is the insufficient knowledge on the relationships between physical, ecological, and valuation variables, and the uncertainty on critical environmental thresholds [34]. The scope of environmental benefit estimations is usually limited to small areas like wetlands, lakes or river reaches, and the range of spatial interactions is narrow. In order to overcome these limitations, environmental benefits should be estimated on the basis of biophysical processes covering most river reaches at basin scale.

The aim of this paper is to analyze the economic and environmental impacts of droughts and water scarcity in the Ebro basin, and the social welfare that can be achieved under alternative water allocation policies. The main objective is to better understand the interactions between environmental and human water uses under water scarcity and drought conditions, by explicitly accounting for environmental benefits linked to river ecological status.

The contribution of this paper over previous literature is the inclusion of the benefits of environmental flows supporting ecosystems in decision making. This is an advance in hydroeconomic modeling that has not been fully developed in earlier studies, because of the difficulties in incorporating environmental components. The innovation over previous hydroeconomic modeling is the calculation of environmental benefits in river reaches, using biophysical information that relates stream flows and ecosystem status. Then, the environmental benefits of river reaches are integrated at basin scale in a framework that accounts for the spatial and sectorial tradeoffs of water allocation, including environmental flows.

The analysis in this study is based on the development of a hydroeconomic model with three components: a reduced-form hydrological component, a regional economy component, and an environmental component. The economy component includes the main urban and irrigation water uses, and the environmental component includes the ecosystem health and the associated environmental benefits. The novelty of this paper lies in the modeling of the environmental component, in which the ecological status response to stream flows is represented using information from biophysical studies relating the flow in river tracts with

ecosystem health. The ecological status is an indicator of the potential of ecosystems to provide goods and services, and it proxies the environmental benefits received by society.

Selected water allocation policies have been evaluated to deal with water scarcity: i) the current institutional cooperation, based in proportional allocation between irrigation districts; ii) environmental institutional cooperation, where proportional allocation is coupled with increased environmental flows to maximize social welfare; iii) water markets, which maximize private benefits; and iv) environmental water markets, where users and the environment exchange water to maximize social welfare. This policy selection follows the approach suggested by Kahil et al. [35]. Institutional cooperation is the current allocation mechanism based on collective action by stakeholders, rather than on administrative coercion or monetary incentives (e.g. pricing).

The paper is structured as follows. The next section describes the Ebro basin, and the following section explains the methodology, outlining the linkages between ecosystem status and environmental benefits, the response of ecosystems to stream flows, the hydroeconomic model framework, and the policy scenarios. Section four presents the results and discussion, and section five concludes with the main findings and policy implications.

## The Ebro basin

The Ebro basin is located in the north-east part of the Iberian Peninsula. It covers an area of 85,600 km$^2$ and supports the economic activities of 3.2 million inhabitants. The Ebro basin is one of the main Mediterranean basins in Europe, containing almost 20 percent of the Spanish territory (S2 Fig). The Ebro basin stream flows sustain 25 percent of both irrigated cropland and hydropower production in the country. Renewable water resources amount to 14,600 million cubic meters (Mm$^3$) per year. Water withdrawals amount to 8,460 Mm$^3$, of which 8,110 Mm$^3$ are surface water diversions and 350 Mm$^3$ groundwater extractions [36]. Water withdrawals for agriculture are 7,680 Mm$^3$ covering 700,000 hectares of irrigated crops. Water abstractions in urban systems amount to 630 Mm$^3$ and direct industry abstractions are nearly 150 Mm$^3$. Non-consumptive water withdrawals are used for cooling thermal power plants (3,100 Mm$^3$) and for hydropower production (38,000 Mm$^3$). Water for agriculture represents 90% of consumptive water demand, and the main irrigated crops are corn, barley, alfalfa, wheat and fruit trees.

Red Natura 2000 spaces are special protection areas for habitat and species, covering 26,000 km$^2$ in the Ebro basin [36]. Water management has been adapted to the environmental regulations governing these protected areas, and environmental flows are maintained even under exceptional conditions during droughts. Environmental flows are included in water planning to achieve good status in different water bodies.

The Confederación Hidrográfica del Ebro (CHE—the Ebro Water Authority) is the institution responsible for managing water in the basin. A special characteristic is the crucial role played by user groups, following the traditional culture of cooperation in the country. The water authority includes representatives from every sector (irrigation, urban, industrial and hydropower), central and state governments, municipalities, farmers' unions, environmental associations, business associations and workers unions.

The CHE is responsible for preparing the water plan of the Ebro, with the objectives of meeting water demand, contributing to regional development, and protecting ecosystems. Ecosystems are protected by setting minimum environmental flows in each river reach. In recent years, there has been a conflict between upstream and downstream states in the basin for the regulation of environmental flows at the Ebro mouth [37].

Environmental flows are based on information from hydrological studies and habitat studies of fish species. Hydrological studies analyze aspects such as flow seasonality, and the rate of

change or river continuity. Habitat studies are weighted usable area (WUA) studies relating the potential habitat for a species with the water flow, from which minimum environmental flows are selected. The Ebro Basin Authority has used a WUA study [38] covering 63 river reaches in order to define environmental flows for each river section.

The above cited WUA study has been used for the estimation of the environmental benefits of river reaches, together with the VANE study [39] which provides estimations of the value of environmental services in the Ebro basin (420 €/ha on average for the whole basin area). The benefits of economic activities are calculated for irrigated cropland and urban use. Data on crop yields, prices, crop water requirements, production costs, availability of water resources, land and labor, biophysical parameters, together with information on urban water use, have been obtained from statistical databases, reports, previous studies and expert consultation [40–47].

## Methods

Interactions between economic activities and the health of ecosystems are driven by multiple and complex biophysical processes, which determine the impacts of economic activities on nature [34]. Harmful alterations of biophysical conditions diminish the services and benefits provided to humans [48]. Both the ecological response to biophysical conditions and the valuation of ecosystem goods and services need to be determined for an adequate representation of environmental benefits [49].

Hydrological regime alterations are driven by the construction of dams for water withdrawals and flood control [50], and these activities reduce stream flows and modify the river morphology [51–54]. Indicators for analyzing biodiversity are used to show the consequences of alterations in the hydrological regime [55]. The ecological response can be studied at population, community, and ecosystem levels [50], although population and community indicators are only partial and do not show the health status of the entire ecosystem [56]. The specification and estimation of the ecological response are challenging tasks [57], and different techniques for the assessment of the ecosystem status are used [58–63].

Ecosystem services are defined as the benefits provided by nature to humans [64], or else as the ecosystem functions that directly and indirectly benefit humans [65]. The classifications of ecosystem services are grouped into four categories: provision, regulation, habitat, and culture and recreation [2,65–67]. The valuation of ecosystems' goods and services is needed for calculating environmental benefits, although specific values of services by type and location remain an unsettled question. Economic valuations are mostly dependent on revealed or stated willingness to pay approaches, based on individual preferences. Valuation results are quite disparate and largely debated, however total economic valuation from use and no-use values is the approach broadly accepted to estimate the value of ecosystems [68].

### Response of ecosystems to stream flow levels

The approaches for establishing environmental flows are mostly based on hydrology, physical habitat simulation, or flow-ecology relationships [9], and environmental flow methodologies are classified in hydrological, hydraulic rating, habitat simulation, and holistic methods [3]. Here we use the habitat simulation method, where habitat suitability is linked to habitat variables such as water velocity, river depth and riverbed composition. The suitability values are then assigned to the area of river reaches to determine the weighted usable area (WUA).

WUA is a measure of the habitat potential to host a specific species given the river streamflow. This methodology accounts for the hydrological, hydraulic (physical and mechanical properties), and biological relationships in order to evaluate environmental flow requirements

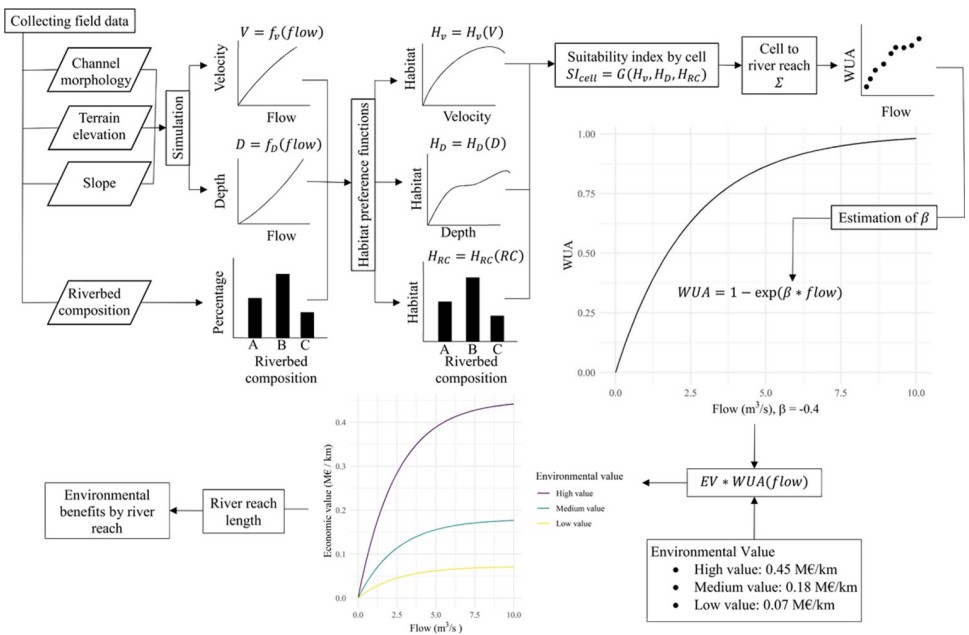

**Fig 1. Environmental benefit response using the weighted usable area.**

[69–71]. Physical habitat simulation requires collecting data on the shape of the river channel, slope of the terrain and riverbed composition for modeling changes in water velocity and river depth with discharge. The habitat preference functions of species indicate the probability of use of a river area under certain conditions, usually water velocity, river depth or riverbed composition, but also water temperature. Preference curves have been modeled extensively for several species and there are numerous examples in the literature (e.g., Grossman and de Sostoa, 1994 [72]; Martínez-Capel and Garcia de Jalón, 1999 [73]).

A river reach is divided in cells, where water velocity and river depth are simulated for levels of streamflow. The results of the simulations and riverbed composition by cell are evaluated with indexes in the habitat preference function, which relates streamflow and habitat adequacy. WUA is then the result of the sum of the suitability habitat index weighted by the size of the cell over the total area of the river reach (Fig 1). The WUA curves represent the habitat's potential to host some particular species, although they are not a predictor of the quantity of fish [69].

Setting up environmental flows in river tracts is burdensome because of the difficulties to obtain habitat-flow relationships with costly field studies. This is solved by using methods that extrapolate available habitat-flow relationship from one part of the river to other parts of the river. These methods estimate the parameters from a functional approach that links WUA and flow. Many functional forms are used in these methods, for example quadratic functions and exponential functions [74,75]. Based on functions used in these studies, the functional form chosen to characterise ecosystem status is the following exponential form:

$$WUA_s(X_s) = 1 - e^{\beta_s X_i}$$

where $WUA_s$ is the weighted usable area in river reach $s$, $X_s$ is the flow in the reach, and $\beta_s$ is the parameter characterizing the WUA response in reach $s$. This function provides an index of the health status of ecosystems. Parameter $\beta$ must be negative and the function meets the

following conditions: the values of the function range from zero to one; the function is strictly increasing; and *WUA* is zero when flow is zero and approaches one as flow rises.

The procedure we followed in the Ebro basin has been to use ecosystem habitat as an indicator of ecosystem status, which is used to define the relationship between flow regime and ecosystem health. Then, we use the economic valuation of services provided by ecosystem health levels, in order to calculate environmental benefits in each river reach (Fig 1). We have used information from several institutions and projects that conduct studies on the valuation of ecosystem services related to water [65,39,76,77].

## Modeling framework of the Ebro basin

The hydroeconomic model of the Ebro basin integrates hydrological, economic, environmental and institutional aspects. The model includes a reduced-form hydrological component, a regional economy component, and an environmental benefit component.

**Reduced-form hydrological component.** The hydrological component is represented by a "reduced-form" model, where the complex hydrological relationships are simplified using historical data and network topology from existing hydrologic models. This is a quick and credible procedure to build a reduced form hydrological model of the studied river basin [78]. The hydrological component represents flows between supply and demand nodes using the hydrological principles of water mass balance and flow continuity. The hydrological component shows the spatial distribution of water flows used by economic sectors and environmental flows (Fig 2). The mathematical formulation is as follows:

$$W_{out_d} = W_{in_d} - W_{loss_d} - Div_d^{IR} - Div_d^{URB} \qquad (1)$$

$$W_{in_{d+1}} = W_{out_d} + r_d^{RI} \cdot (Div_d^{IR}) + r_d^{URB} \cdot (Div_d^{URB}) + RO_{d+1} \qquad (2)$$

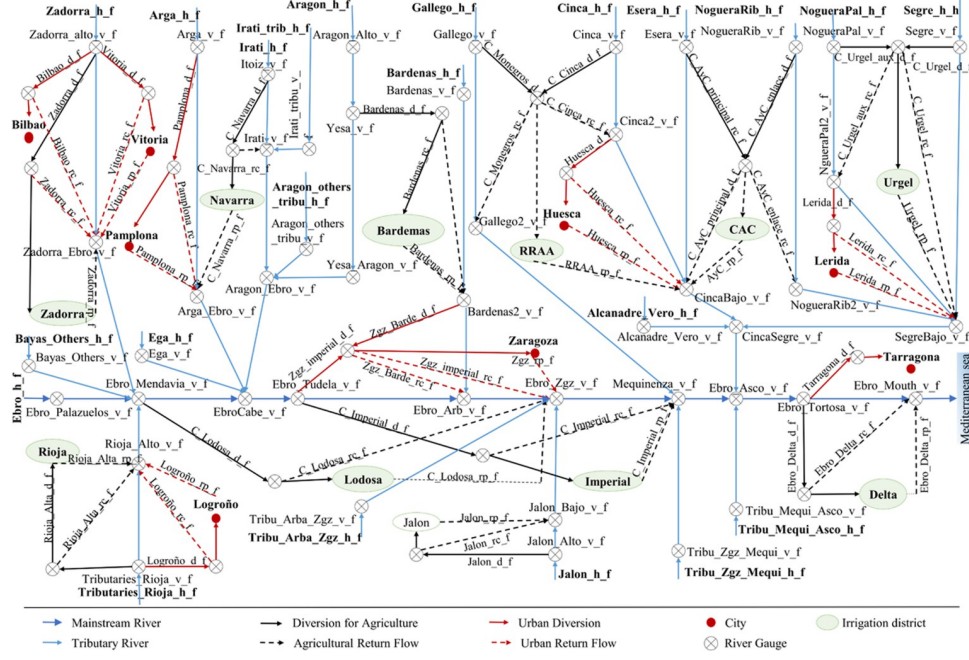

**Fig 2. Hydrological network of the Ebro basin.**

$$W_{out_{mouth}} \geq E_{mouth}^{min} \tag{3}$$

Eq (1) is the mass balance equation, and it determines water outflow $W_{out_d}$ in river reach $d$, which is equal to water inflow $W_{in_d}$, minus water losses $W_{loss_d}$, water abstraction for irrigation $Div_d^{IR}$, and abstraction for urban and industrial use $Div_d^{URB}$. Eq (2) guarantees river flow continuity, in which water inflow in the following river reach $W_{in_{d+1}}$ is the sum of the water outflow from the previous reach $W_{out_d}$, return flows from previous irrigation districts $[r_d^{RI} \cdot (Div_d^{IR})]$, urban return flows $[r_d^{URB} \cdot (Div_d^{URB})]$, and the flow entering this river reach from tributaries $RO_{d+1}$. Eq (3) states that water outflow at the mouth of the Ebro $W_{out_{mouth}}$ must be greater than or equal to the minimum environmental flow in the river reach. Further details on the hydrological component can be found in the GAMS code of the model in Supplementary Materials.

The hydrological component has been calibrated adjusting the model parameters by introducing auxiliary variables for every river reach, in order to reproduce the observed system states of nature such as stream flows under baseline conditions. Calibration is used to close the mass balance equation, since there are water inflows and outflows in the system that cannot be observed (for example, underground flows, evaporation or some return flows). Calibration includes non-observed flows, which are the difference between flows estimated with the model and flows measured at gauging stations.

**Regional economic component.** The regional economic component includes agricultural irrigation and urban water use. There is a model for agricultural activities in every irrigation district, where farmers' private benefits from crop production are constrained by technical and resource restrictions. Crop yield functions are assumed linear and decreasing, and output and input prices are constant. These irrigation benefits enter into the objective function of the integrated model (Eq 16), which is maximized. The formulation is the following:

$$B_k^{IR} = \sum_{ij} C'_{ijk} X_{ijk} \tag{4}$$

s.t.

$$\sum_i X_{ijk} \leq T_{land_{kj}}; \; j = flood, \; sprinkler, \; drip \tag{5}$$

$$\sum_{ij} W_{ijk} X_{ijk} <= T_{water_k} \tag{6}$$

$$\sum_{ij} M_{ijk} X_{ijk} \leq T_{la_k} \tag{7}$$

$$X_{ijk} \geq 0 \tag{8}$$

where $B_k^{IR}$ is private benefit in irrigation district $k$, and $C'_{ijk}$ is net income of crop $i$ using irrigation technology $j$. The decision variable of the problem is $X_{ijk}$, which is acreage of crop $i$ under irrigation technology $j$. Eq (5) represents the restriction of available land $T_{land_{kj}}$ in irrigation district $k$ equipped with irrigation system $j$. The water available $T_{water_k}$ in irrigation district $k$ is given by Eq (6), where $W_{ijk}$ is the water requirement of crop $i$ with technology $j$. The water available $T_{water_k}$ is the variable linking the optimization model of irrigation districts and the hydrological component. The labor constraint (7) represents labor availability $T_{la_k}$ in irrigation district $k$, where $M_{ijk}$ is the labor requirement of crop $i$ with irrigation system $j$.

This irrigation model includes the major crops in every irrigation district. Irrigation systems for field crops are flood and sprinkler, and for fruit trees and vegetables are drip and flood. Net income per hectare $C'_{ijk}$ is the difference between crop revenue and direct and indirect costs (including capital amortization) and it is expressed by $C'_{ijk} = P_i Y_{ijk} - CP_i$ where $P_i$ is price of crop $i$, $Y_{ijk}$ is yield of crop $i$ under technology $j$ in the irrigation district $k$, and $CP_i$ are direct and indirect costs of crop $i$ (including water costs).

The crop yield function is linear and represents a decreasing crop yield when additional land is assigned to crop production, based on the principle of Ricardian rent. The first lands in production have the highest yields, and yields fall off as less-suitable lands enter production. The crop function relates yields with acreage of crop $i$ under irrigation technology $j$, and is defined as:

$$Y_{ijk} = \beta_{0_{ijk}} + \beta_{1ijk} X_{ijk} \tag{9}$$

The agricultural component is calibrated using Positive Mathematical Programming (PMP) to reproduce the observed land and water use under baseline conditions, and to address the problem of crop overspecialization [79]. Calibration follows the PMP procedure by [80], where parameters are estimated for a linear yield function [Eq (9)] based on the first-order conditions of benefit maximization.

The modeling of urban water calculates economic surplus, the sum of consumer and producer surpluses in the basin's main cities. The urban economic surplus enters the objective function of the integrated model (Eq 16), which is maximized. The formulation of the urban sector is expressed by:

$$B_u^{URB} = \left(a_{du} Q_{du} - \tfrac{1}{2} b_{du} Q_{du}^2 - a_{su} Q_{su} - \tfrac{1}{2} b_{su} Q_{su}^2\right) \tag{10}$$

s.t.

$$Q_{du} - Q_{su} \leq 0 \tag{11}$$

$$Q_{du};\ Q_{su} \geq 0 \tag{12}$$

where $B_u^{URB}$ is the consumer and producer surplus in city $u$. The variables $Q_{su}$ and $Q_{du}$ are water supply and demand in city $u$, respectively. The parameters $a_{du}$ and $b_{du}$ are the constant term and the slope of the inverse demand function, and the parameters $a_{su}$ and $b_{su}$ are the constant term and the slope of the water supply function. Eq (11) states that the supply must be equal to or greater than the demand for water. The water supply $Q_{su}$ is the variable linking urban water with the hydrological component. The equation parameters have been obtained from the studies by Arbués et al. [46] and Arbués et al. [47].

**Environmental component.** The environmental benefits of aquatic ecosystem in the basin depend on the health status of ecosystems, where the relationship between the river's habitat status and stream flows is expressed by the WUA. A study of the WUA in the Ebro Basin provides data for these relationships for every section in the basin, based on their hydrological characteristics. The relationships are estimated by an exponential function calibrated to the data from the WUA study [38].

The WUA study covers 14 segments of the hydrology network of the Ebro Basin, characterizing the ecological status in every river reach. The benefits aquatic ecosystems generate are

given by the following expressions:

$$WUA_s(W_s) = 1 - e^{\beta_s W_s} \qquad (13)$$

$$EW_s^{eco} = WUA_s(W_s) \qquad (14)$$

$$B_s^{eco} = VE \cdot l_s \cdot EW_s^{eco} \qquad (15)$$

where $WUA_s$ is the weighted usable area in river reach $s$, $W_s$ is average flow in $s$, and $EW_s^{eco}$ is the health status of ecosystem $eco$ in $s$. The weighted usable area $WUA_s$ depends on average flow $W_s$ and the estimated parameter $\beta_s$, considering the months with less water availability. The parameter $\beta_s$ has been estimated through non-linear regression for 14 of the 63 locations where information is available (Table 1).

WUA studies require collecting data of the river waterbed composition and undertaking a topographic study of the river reach. The information on the morphology and topology of the terrain enables the simulations of the hydraulic variables, water velocity and water depth, generated by a specific streamflow. This information is combined with habitat studies that describe fish preferences to water velocity, water depth and waterbed characteristics, in order to calculate the habitat potential. The WUA is the habitat potential weighted by the total area of the river reach. The studies of the Ebro basin authority includes 64 river reaches of the Ebro basin. For each river reach, a representative fish species is selected and the WUA is obtained for three different life-stages: fray, juvenile and adult. WUA for a fish species for the dry and wet seasons are produced combining life-stage WUA results and the mean flow at the river reach.

This relationship and its corresponding parameter for each river reach determines the habitat size and the ecosystem potential to contain species given a specific flow. The benefit $B_s^{eco}$ of ecosystems in reach $s$ depends on the ecosystem health status $EW_s^{eco}$ (from zero to one), the length $l_s$ of the reach (km), and the economic value $VE_s$ of ecosystem services (€/km) (Eq 15).

The economic value of ecosystem services $VE_s$ is derived from published studies in the literature. Values are usually given in euros per hectare of riverbed, which can be converted to euros per kilometer by knowing the surface area of the river reach covered by water and the length of the river reach. Valuation in the literature ranges from 2,000 to 40,000 €/ha of riverbed covered by water [65,76,77], and from this range we select an average value of 24,000 €/ha for ecosystems' services in the Ebro. The area covered by water in the rivers of the Ebro basin is 68,000 ha with a total length of 8,900 km [81], therefore the average value in euros per kilometer is 0.180 M€/km (24,000 €/ha•68,000 ha/8,900 km). However, ecosystems' values are spatially heterogeneous in the basin, with values higher for mountain rivers than for streams in

**Table 1. Results of the WUA regression for 14 river reaches in the Ebro.**

$WUA_{river\ reach} = 1 - e^{\beta_{river\ reach}*flow_{river\ reach}}$

| River reach ID | n | β | Standard error | t | Pr(>\|t\|) | River reach ID | n | β | Standard error | t | Pr(>\|t\|) |
|---|---|---|---|---|---|---|---|---|---|---|---|
| 202 | 33 | -9.65 | 1.82 | -5.31 | < 0.01 | 428 | 34 | -0.30 | 0.03 | -12.01 | < 0.01 |
| 264 | 32 | -2.07 | 0.21 | -9.84 | < 0.01 | 433 | 44 | -0.10 | 0.01 | -12.05 | < 0.01 |
| 274 | 24 | -1.42 | 0.17 | -8.52 | < 0.01 | 441 | 46 | -8.14 | 2.75 | -2.96 | < 0.01 |
| 406 | 41 | -16.39 | 2.74 | -5.98 | < 0.01 | 446 | 20 | -1.96 | 0.31 | -6.44 | < 0.01 |
| 418 | 39 | -1.89 | 0.22 | -8.74 | < 0.01 | 455 | 26 | -9.61 | 2.53 | -3.79 | < 0.01 |
| 421 | 41 | -0.12 | 0.01 | -13.54 | < 0.01 | 463 | 30 | -0.23 | 0.05 | -4.39 | < 0.01 |
| 426 | 24 | -0.52 | 0.06 | -7.95 | < 0.01 | 662 | 40 | -0.26 | 0.02 | -14.60 | < 0.01 |

the valley as estimated by MARM [39]. Following the valuation ranges in the literature, three valuation levels are chosen: a low value (0.072 M€/km) for river sections with moderate environmental value in the main stem of the Ebro and in some right bank tributaries, a medium value (0.180 M€/km) for non-mountain Ebro tributaries, and a high value (0.450 M€/km) for mountain river reaches and also for the Ebro mouth where the Ebro Delta is located.

## Modeling policy scenarios

The optimization model integrates the hydrologic, economic and environmental components. The model maximizes the basin's benefits subject to hydrological, technical, resource and institutional constraints in every economic sector and location. The optimization problem is defined by the expression:

$$Max \sum_k B_k^{ir} + \sum_u B_u^{urb} + \sum_s B_s^{eco} \tag{16}$$

subject to Eqs (1)–(15) and the constraints of water availability in the basin:

$$Div_d^l \leq W_{in_d} \ \forall \, l, d \quad \text{where } l = k, u \tag{17}$$

$$\sum_d Div_d^l \leq \overline{W} \ \forall \, l = k, u \tag{18}$$

where $B_k^{ri}$ are the benefits of each irrigation district, $B_u^{urb}$ are the befits of each urban center, and $B_s^{eco}$ are the environmental benefits of each river reach. Eqs (17) and (18) allocate water among uses and locations. Eq (17) ensures that water extractions at each demand node are lower than or equal to net water inflows in river reaches. Eq (18) indicate that basin water withdrawals cannot exceed water availability $\overline{W}$ in the basin. The optimization problem has been formulated with the GAMS optimization package, using the CONOPT 4 solver. The GAMS code of the model is available in the Supplementary Materials.

The model is used to analyze the impact of droughts on the basin economic activities and environment. Water inflows into the system under normal weather conditions are 14,600 hm³, corresponding to the average inflows in the 1980–2014 period [36]. Water inflows into the system are reduced by 40% under severe drought conditions. This reduction is chosen by considering the previous four severe droughts with a fall around 40% in basin inflows during the last 30 years (in years 1989, 2002, 2005 and 2012). Climate change will further reduce basin inflows by 12% at the end of this century [82]. The model is also used to analyze the economic and environmental effects of drought management policies. The scenarios are a combination of the drought situation with the following four policies developed to deal with drought.

**Institutional cooperation.**   Under drought conditions, the basin authority reduces water allocations for irrigation in proportion to drought intensity (fall in inflows). Consequently, the water shortfall is shared between irrigation districts, but it also reduces environmental flows. This is the policy currently applied in the Ebro basin. In the model, the water allocations to irrigation districts are reduced in proportion to the reduction in inflows due to drought.

**Environmental institutional cooperation.**   Farmers receive the same allocations than under *institutional cooperation*, but then the basin authority purchases water from farmers for the environment in order to maximize social benefits, the sum of both private and environmental benefits. Water exchanges between irrigation districts are not allowed. In the model, the environmental benefits are included in the objective function, and the basin authority buys water to achieve the optimal solution found for stream flows.

**Water markets.**   Farmers face the reduced water allocations of institutional cooperation, but then these water allocations can be exchanged among irrigation districts, maximizing the

private benefits of water use. There is no direct exchange of water between selling and buying irrigation districts, but rather the selling district reduces withdrawals, and the buying district augments withdrawals in their respective river reaches. In the model, a restriction is introduced that limits the sum of reduced water allocations to districts (allowing trading), instead of limiting each district to its reduced water allocation.

**Environmental water markets.** Same water allocations as in institutional cooperation. Water can be exchanged between irrigation districts, and also the basin authority participates in the water market by acquiring water to protect environmental benefits in river reaches. This policy enhances both private and environmental gains, so it is an appealing policy to capture the private benefits of markets while protecting ecosystems. In the model, the environmental benefits are included in the objective function, and the water authority buys water from irrigation districts to achieve the optimal solution found for stream flows.

The baseline scenario assumes the policy currently applied under normal weather conditions. The minimum environmental streamflow at the mouth of the Ebro is set at 3,000 hm$^3$, and the urban water supply is guaranteed in all scenarios.

## Results

In the *baseline scenario* with normal weather, the social benefits of water from economic activities and the environment are 3,442 M€, of which 629 M€ are from irrigation, 1,857 M€ from urban use, and 956 M€ from the environment. Water withdrawals are 5,380 Mm$^3$ in irrigation and 402 Mm$^3$ in urban use, while environmental flow at the mouth is 8,895 Mm$^3$. This environmental flow collects all water coming from stream flows across the basin, sustaining aquatic ecosystems (Tables 2 and S1). Environmental benefits are displayed by local watershed, which is the water management unit in the basin (Figs 3 and S1).

The irrigated area under normal weather is 529,000 ha, distributed between field crops (75%), fruit trees (20%) and vegetables (5%). Fruit trees and vegetables generate half of farmers' benefits. The cropping pattern is quite different by irrigation district, with Riegos del Alto Aragón and Bardenas specializing in field crops, Riegos del Jalón specializing in fruit trees, and Canal de Lodosa specializing in vegetables.

### Institutional cooperation

Water availability is reduced 40% under severe drought conditions. In the *institutional cooperation* policy, water allocations to irrigation are a share of water inflows into the basin, and they are reduced in proportion to drought intensity. Water withdrawals for urban centers are maintained because of the priority of urban supply over other uses, including the environment. The decrease in the availability of water causes losses in irrigation and environmental benefits, reducing social benefits. During drought periods and under the *institutional cooperation* policy, irrigation withdrawals fall to 3,230 Mm$^3$ (-40%). This reduction leads to less irrigated area (-37%) and private benefits (-26%), compared to the *baseline scenario*. The streamflow at the river mouth drops from 8,895 Mm$^3$ to 5,350 Mm$^3$ under drought (Table 2).

Environmental benefits decrease by 20% during drought, affecting the main stem and the left and right bank tributaries. The tributaries on the left bank provide most of the water in the Ebro, and pressures on environmental flows are mostly from irrigation withdrawals. Water scarcity in summer months determines the ecosystem sensitivity to drought. Under drought, urban benefits are maintained but irrigation and environmental benefits fall.

The *environmental institutional cooperation* policy reallocates water between irrigation and the environment to maximize social wellbeing. Social wellbeing is the sum of private and environmental benefits, minus the public expenses to buy irrigation water for the environment.

**Table 2. Policies under drought conditions: Institutional cooperation, environmental institutional cooperation, water markets, and environmental water markets.**

| Weather scenario | Normal weather | Severe drought | | | |
|---|---|---|---|---|---|
| Policies | Baseline scenario | Institutional cooperation | Environmental institutional cooperation | Water markets | Environmental water markets |
| Water use (Mm$^3$) | | | | | |
| Water use | 5,782 | 3,632 | 3,047 | 3,627 | 3,232 |
| Irrigation | 5,380 | 3,230 | 2,645 | 3,225 | 2,830 |
| Urban | 402 | 402 | 402 | 402 | 402 |
| Water exchanges | | | 585 | 235 | 780 |
| Between irrigators | | | | 235 | 380 |
| Between irrigators and environment | | | 585 | | 400 |
| Environmental flow at mouth | 8,895 | 5,350 | 5,540 | 5,345 | 5,435 |
| Irrigation surface area (1,000 ha) | | | | | |
| Surface area | 529 | 332 | 275 | 348 | 293 |
| Field crops | 400 | 219 | 165 | 229 | 182 |
| Fruit trees | 104 | 93 | 90 | 97 | 90 |
| Vegetables | 25 | 20 | 20 | 22 | 21 |
| Private and environmental benefits (M€) | | | | | |
| Private benefits | 2,486 | 2,321 | 2,332 | 2,340 | 2,346 |
| Irrigation | 629 | 464 | 475 | 483 | 489 |
| Urban | 1,857 | 1,857 | 1,857 | 1,857 | 1,857 |
| Environmental benefits | 956 | 761 | 834 | 719 | 826 |
| Social benefits | 3,442 | 3,082 | 3,105 | 3,059 | 3,118 |

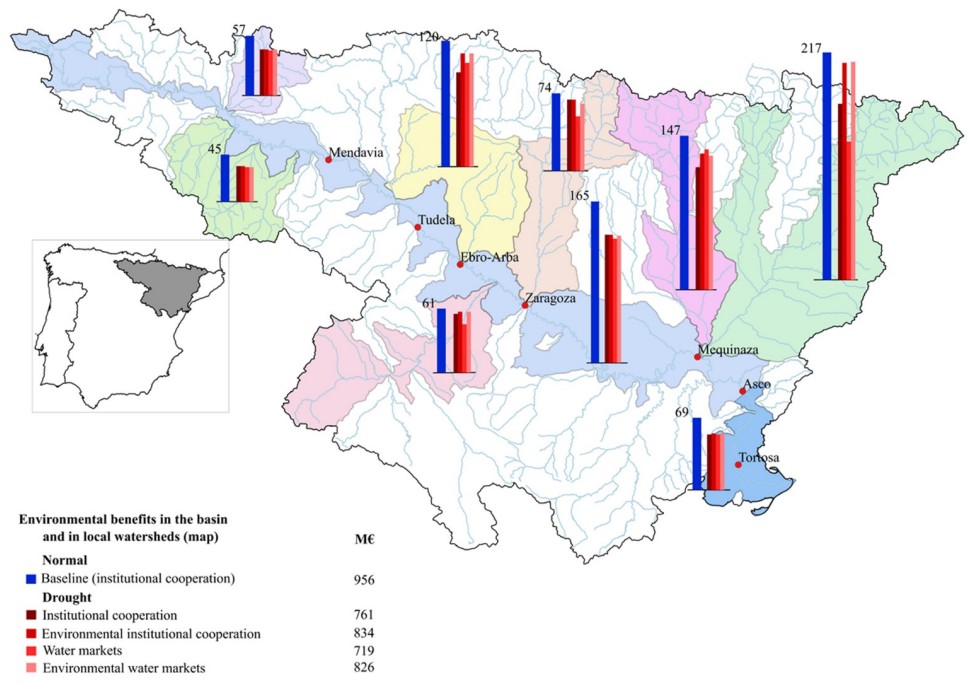

**Fig 3. Environmental benefits of policies under normal and drought conditions.** Reprinted from Confederación Hidrográfica del Ebro under a CC BY license, with permission from Confederación Hidrográfica del Ebro.

Acquiring water for the environment improves ecosystem status, especially in areas with high potential for improvement, while maintaining irrigators' income. Under drought conditions the basin authority purchases 600 Mm³ for €60 M. This reallocation increases environmental benefits by €70 M (9%) and irrigation benefits increase by nearly €9 M.

## Water markets

Under the *water markets* policy, irrigation districts exchange water and maximize their private benefits, but environmental benefits are disregarded. Under drought, water withdrawals are 3,225 Mm³ and cultivated area falls to 348,000 ha, with lower irrigation benefits. Flows at the river mouth are 5,345 Mm³ under drought, well above the minimum environmental flow at 3,000 Mm³. This minimum environmental flow is satisfied by all policy scenarios (Table 2, Fig 4). Irrigation districts exchange 235 Mm³ under drought, where irrigation districts with low water efficiency and specialized in field crops sell water to irrigation districts with efficient irrigation systems and profitable crops. Water exchanges enable a larger cultivated area compared with the institutional cooperation policy. That is why crop water consumption (evapotranspiration) is higher under *water markets*, reducing basin stream flows and environmental benefits. The *water markets* policy generates higher private benefits and lower environmental benefits compared with *institutional cooperation*.

Irrigation districts buying water expand withdrawals and deplete water streams in their river reach, while the opposite takes place in selling districts. Consequently, water exchanges relieve environmental pressures in selling districts and aggravate pressures in buying districts. This can be observed in the Cinca watershed (Fig 3 in pink) where irrigation districts sell water and environmental benefits increase above any other policy, or in the Segre watershed (Fig 3 in green at right) where irrigation districts buy water and environmental benefits fall.

## Environmental water markets

The purpose of this policy is to maximize social benefits by including both the private and public benefits of water, in order to internalize the external effects of water markets. The

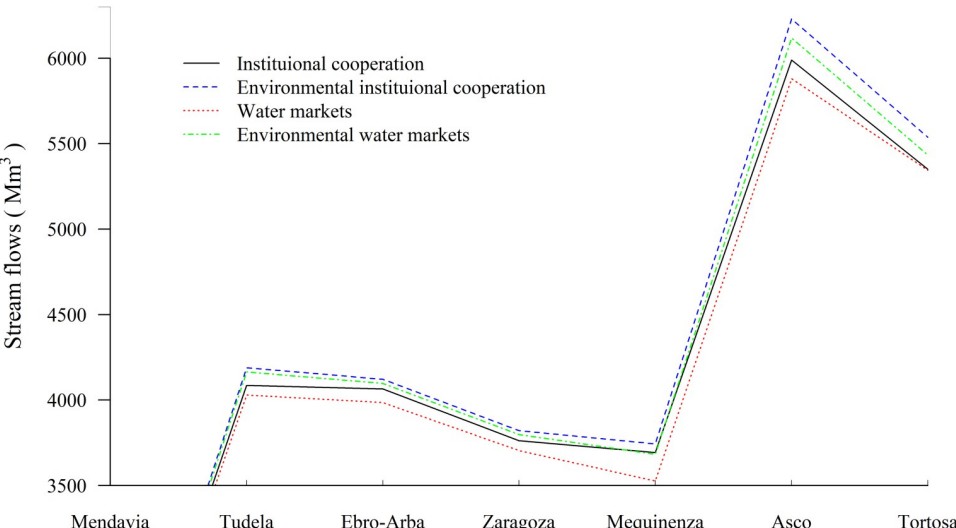

**Fig 4. Stream flows along the Ebro for each policy under drought conditions (Mm³/year).** [Note below the figure] Minimum annual environmental flows, regulated by the Ebro Basin Authority, are: 300 Mm³ at Mendavia under normal and drought conditions; 945 Mm³ at Zaragoza under normal conditions and 640 Mm³ under drought conditions; and 3,000 Mm³ in Tortosa under normal and drought conditions [36].

*environmental water markets* policy consists in water trading not only between irrigation districts but also with the environment, where the basin authority purchases water for the environment. Under drought conditions, water exchanges between irrigation districts are 380 Mm$^3$, and water exchanges between irrigation districts and the environment are 400 Mm$^3$. Irrigation withdrawals, irrigated area and crop production decrease compared with the *water markets* policy. The benefits of irrigation districts are obtained from crop production (393 M €) and from water trading with other districts (42 M€) and the environment (54 M€). The private benefits of irrigation under *environmental water markets* are above those of *water markets*, since crop production benefits and income from water sales are added in environmental water markets. The benefits of *environmental water markets* strictly dominate those from *water markets* in both private and environmental benefits.

## Discussion

*Environmental water markets* and *environmental institutional cooperation* reduce water withdrawals, especially in river sections where environmental sensitivity is high. Water exchanges between irrigation and the environment increases environmental flows by reducing the cultivated area of field crops and fruit trees, while maintaining vegetables. The area of fruit trees decreases in irrigation districts withdrawing from river sections which are more environmentally sensitive, especially on the right bank.

 *Environmental institutional cooperation* achieves a better ecosystem protection than *environmental water markets*, by acquiring almost 50% more water for the environment. Because there are no water exchanges among districts, farmers cannot take advantage of the private gains achieved under *environmental water markets*. Irrigation withdrawals under *environmental water markets* are around 200 Mm$^3$ larger than with *environmental institutional cooperation*, showing the trade-off between private and environmental benefits of these policies. *Environmental institutional cooperation* provides more environmental flows and better ecological status in all river reaches across the basin. In contrast, *environmental water markets* improve the ecological status in some river reaches at the expense of others reaches.

 The spatial location of irrigation districts, the relationship between available water and water withdrawals, ecosystem sensitivity to water scarcity, crop patterns and irrigation technologies are the factors driving the impact caused by water scarcity on economic activities and the environment. Environmental and irrigation responses depend on spatial location, because water available and water withdrawal intensity are heterogeneous throughout the basin. Ecosystem sensitivity, the economic value of the ecosystem services, and alternative uses of water shape environmental flows. Under drought, each policy results in different distribution of losses between private and environmental benefits and the ensuing trade-offs.

 During droughts, *institutional cooperation* distributes losses to irrigation districts and the environment in proportion to water allocations in normal years. *Environmental institutional cooperation* enhances social benefits providing additional protection to the environment. *Water markets* maximize private benefits of irrigation but disregards environmental benefits, and *environmental water markets* deliver both private and environmental gains, capturing the private benefits of markets while protecting ecosystems. Water exchanges from irrigation to the environment enhances environmental benefits in the river reach of the exchange, and also in downstream river reaches, thus boosting the exchange benefits.

 Reponses to drought entail maintaining high profitable crops and efficient irrigation systems, and therefore water scarcity has greater negative impacts on cultivation of less technically-advanced irrigation districts specialized in field crops. These irrigation districts with field

crops and low water efficiency sell water to efficient and profitable irrigation districts and also to the environment.

The *water markets* policy provides the lowest social benefits because environmental losses overcome irrigation gains. The *environmental water markets* policy accounts for the market shortcomings by internalizing the negative external effects of economic activities, enhancing social welfare. However, the ecological status with this "environmental" trading improves in some river reaches but worsens in others. Equity considerations are better addressed by the *institutional cooperation* policy, which distributes proportionally the drought water scarcity among all irrigation districts and aquatic ecosystems in the basin. This policy not only protects the environment but also contributes to a more equitable distribution of water and benefits. *Environmental institutional cooperation* further enhances social benefits by a full consideration of environmental benefits into water allocation decisions. The tradeoff between environmental and private benefits is obtained by comparing the outcomes of benefits under each policy. For example, there are gains in environmental benefits and losses in private benefits when *environmental institutional cooperation* is compared with *water markets*.

The range of the increase in environmental benefits between *environmental institutional cooperation* (high flow protection) and *water markets* (low flow protection) is quite close (115 M€ = 834–719, Table 1) to the environmental damages from water uses estimated by Garcia de Jalon et al. [83] in northern Spain. They estimate the environmental costs of streamflow variability induced by human extractions, applying the "polluter pays" principle. Their estimation of damages is 0.02 €/m$^3$, which in the case of the Ebro will amount to 107 M€ (0.02 €/m$^3$·5,345·10$^6$ m$^3$). This similarity in the estimation of environmental damages in the two studies, that use quite different methodologies, strengthen the reliability of the damages found in this study.

Previous studies on environmental benefits using hydroeconomic modeling are only partial and provide only rough estimations. Ringler and Cai [21] use two functional forms for fisheries and wetlands in the Mekong River, however the authors acknowledge that their evaluation requires further improvement. The study by Grossmann and Dietrich [26] only analyzes the Spreewald wetland in the mid-reaches of the Spree River, but not the environmental benefits in the whole basin. Bekchanov et al. [27] analyze investments in irrigation technologies to improve irrigation, hydropower and ecosystem benefits in the Aral Sea Basin. They estimate ecosystem benefits only in the river delta, and ecosystem benefits are a simple linear function of flows at the mouth of the Amur Daria and Syr Daria rivers, recognizing that these benefits are only rough estimates.

The wider implications of the approach taken here are that: i) the importance for decision making of considering environmental benefits for whole river basins, rather than for partial locations that preclude finding the efficient allocation of water in the basin; ii) the advance of estimating environmental benefits using the weighted usable area, based on observed field data for every river reach along the hydrological basin network; iii) the assessment of drought policies in this study provide significant information on the trade-offs faced by decision makers in order to balance private benefits from economic activities and public benefits from ecosystem protection.

The modeling approach taken in our study present several limitations. One limitation is that the model is static, and the model can be converted to dynamic by including the reservoirs in the hydrological network of the basin. Also, more detailed findings can be obtained by changing the data step from yearly to monthly, because stream flows are predominantly threatened in summer months during droughts. Another aspect deserving improvement is the calculation of environmental benefits in river reaches. This requires a more precise valuation of the ecosystem services provided by river reaches with specific studies on local watersheds in the

basin, and also better information on the response of ecosystem health status to stream flows, including more species and habitats in the weighted usable area technique for assessing environmental flows.

## Conclusions and policy implications

This paper develops a hydroeconomic model that is used to analyze water allocation policies in the Ebro basin under water scarcity. The contribution made by this study is the inclusion of an environmental component linking stream flows with the status of aquatic ecosystems and their environmental benefits. Most hydroeconomic models consider environmental protection by setting minimum flow requirements, rather than representing the complex ecological responses to stream flows [34,84].

The model is used to analyze water allocation policies in the Ebro basin, a basin with several semi-arid regions where droughts trigger water scarcity problems and significant impacts on irrigation and the environment. This problem is common to arid and semi-arid basins worldwide, and it is expected to worsen as a consequence of climate change and pressures from economic and population growth. The methodology and results of this policy analysis could be applied to other basins facing similar problems of mounting withdrawals and ecosystems degradation.

The hydroeconomic model includes the benefits from irrigated agriculture, urban use, and environmental flows. The purpose is to encompass both the spatial and sectoral interrelationships of water allocations in the basin, in order to find the economic and environmental impacts of drought and the social welfare that can be achieved under alternative water allocation policies.

The impacts of drought and the ensuing social welfare under the policies considered provide important information to support decision making. The *water markets* policy generates higher private benefits than the current *institutional cooperation* policy, but attains lower environmental benefits. The public good characteristics of environmental flows imply that ecosystem services are external to markets, and the market failure has to be corrected. One alternative is the *environmental water markets* policy, where the basin authority buys water for the river. This water sharing between economic sectors and the environment is based on the corresponding private and environmental benefits. Results from this "environmental" trading show that the policy of *water markets* is strictly dominated by *environmental water markets*, because the latter provides gains for both farmers and the environment, thus achieving higher social benefits. However, under this policy the ecosystem status improves in some river reaches but worsens in other reaches. *Environmental water markets* require also stakeholders' cooperation as an essential ingredient to curtail losses from water scarcity, and achieve the "good ecological status" of water bodies promoted by European legislation [85–92]. Well functioning *environmental water markets* would enhance social benefits [89–92], but this entails the support of strong institutions fostering collective action [29], which would prevent third party effects that mostly affect the environment.

In contrast, the *environmental institutional cooperation* policy achieves a more equitably distribution of drought shortfalls among irrigation districts, and attains the highest environmental protection. The water authority approach is based on stakeholders' cooperation in water management at local watershed and basin levels. This collaboration among water users, administrations and other stakeholders is essential for implementing sustainable water management.

The design and implementation of sustainable basin management requires information on the response of ecosystems to stream flows, and on the valuation of ecosystems services. This

involves multidisciplinary research with considerable efforts in terms of resources and time, and this research is lacking at present in most basins around the world. Such information is also needed to gain the support of stakeholders in reversing the current global degradation of water resources.

The challenge for water management is balancing the effects of water allocations between economic activities and the environment in decision-making. Market policies overlook the externalities from private water usage, and public interventions are needed to deal with market failures. But even with institutional cooperation in the case of Spain, the situation in basins shows that basin stream flows have gradually decreased in recent decades, and more virtuous collective action outcomes are needed to prevent further deterioration of stream flows. In many basins around the world, the decline in environmental flows is damaging aquatic ecosystems, with private benefits increasing in the short run at the expense of environmental benefits. But in the long run the degradation of hydrological systems would become unsustainable, with strong negative impacts on both economic activities and ecosystems.

## Supporting information

**S1 Fig. Environmental benefits of policies under normal weather, and moderate and severe drought[a].** Reprinted from Confederación Hidrográfica del Ebro under a CC BY license, with permission from Confederación Hidrográfica del Ebro. [a] We present here additional results to the impacts of severe drought (-40% inflow reduction) considered in the main text of the article. These results correspond to a moderate drought scenario, where the reduction of inflows is lowered to 30% applying the same methodology.
(JPG)

**S2 Fig. Map of irrigation districts and urban centers in the Ebro basin.** Reprinted from Confederación Hidrográfica del Ebro under a CC BY license, with permission from Confederación Hidrográfica del Ebro.
(JPG)

**S1 Table. Policies under drought conditions: Institutional cooperation, environmental institutional cooperation, water markets, and environmental water markets.** [a] We present here additional results to the impacts of severe drought (-40% inflow reduction) considered in the main text of the article. These results correspond to a moderate drought scenario, where the reduction of inflows is lowered to 30% applying the same methodology.
(DOCX)

**S1 File.**
(GMS)

## Acknowledgments

Special assistance has been provided by María Ángeles Lorenzo and Daniel Isidoro (CITA-DGA), and by Rogelio Galván and Miguel Ángel García Vera (CHE-MITECO). Part of the work by Daniel Crespo was developed at the School of Public Policy at UC Riverside, where he received advice from Dr. Darrel Jenerette, Dr. Louis Santiago, Dr. Kurt E. Anderson and Dr. Sergio Rey. Daniel Crespo has conducted this study with the support of a PhD grant from the Spanish Ministry for Science and Innovation.

## Author Contributions

**Conceptualization:** Daniel Crespo, Jose Albiac, Ariel Dinar, Taher Kahil.

**Data curation:** Daniel Crespo, Jose Albiac.

**Formal analysis:** Daniel Crespo, Jose Albiac, Taher Kahil.

**Funding acquisition:** Jose Albiac.

**Investigation:** Daniel Crespo, Jose Albiac, Ariel Dinar, Encarna Esteban, Taher Kahil.

**Methodology:** Daniel Crespo, Jose Albiac, Ariel Dinar, Encarna Esteban, Taher Kahil.

**Project administration:** Jose Albiac.

**Supervision:** Jose Albiac, Taher Kahil.

**Validation:** Jose Albiac, Taher Kahil.

**Writing – original draft:** Daniel Crespo, Jose Albiac.

**Writing – review & editing:** Jose Albiac, Ariel Dinar, Encarna Esteban, Taher Kahil.

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
