## [Decision Letter · Decision Letter 0]

1 Dec 2021

PONE-D-21-31923Integrating ecosystem benefits for sustainable water allocation in hydroeconomic modelingPLOS ONE

Dear Dr. Albiac,

Thank you for submitting your manuscript to PLOS ONE. After careful consideration, we feel that it has merit but does not fully meet PLOS ONE’s publication criteria as it currently stands. Therefore, we invite you to submit a revised version of the manuscript that addresses the points raised during the review process.

The work reported in the manuscript is novel and contributes to an important area of research. Both reviewers agree but identify some problems with the way the work is presented. Please carefully revise the manuscript according to the reviewer recommendations. Note that one of the reviewers provides additional comments in an annotated version as an attachment.

We look forward to receiving your revised manuscript.

Kind regards,

Neville Crossman, Ph.D.

Academic Editor

PLOS ONE

Journal Requirements:

"This study has been financed by the projects INIA RTA2014-00050-00-00 and INIA RTA2017-00082-00-00 of the Ministry for Science and Innovation, partly financed by European ERDF funds, and by support received from the ECONATURA research group of the Government of Aragon. Daniel Crespo has conducted this study with the support of a PhD grant from INIA. Part of the work by Daniel Crespo was developed at the School of Public Policy at UC Riverside."

"J.A received the funding from projects INIA RTA2014-00050-00-00 and INIA RTA2017-00082-00-00 of the Ministry for Science and Innovation, partly financed by European ERDF funds (https://www.ciencia.gob.es/site-web/).

4. We note that Figures 2 and S1 in your submission contain [map/satellite] images which may be copyrighted. All PLOS content is published under the Creative Commons Attribution License (CC BY 4.0), which means that the manuscript, images, and Supporting Information files will be freely available online, and any third party is permitted to access, download, copy, distribute, and use these materials in any way, even commercially, with proper attribution. For these reasons, we cannot publish previously copyrighted maps or satellite images created using proprietary data, such as Google software (Google Maps, Street View, and Earth). For more information, see our copyright guidelines: http://journals.plos.org/plosone/s/licenses-and-copyright.

a. You may seek permission from the original copyright holder of Figures 2 and S1 to publish the content specifically under the CC BY 4.0 license.  

Reviewers' comments:

Reviewer's Responses to Questions

**Comments to the Author**

1. Is the manuscript technically sound, and do the data support the conclusions?

Reviewer #1: Partly

Reviewer #2: Partly

2. Has the statistical analysis been performed appropriately and rigorously? 

Reviewer #1: I Don't Know

Reviewer #2: N/A

3. Have the authors made all data underlying the findings in their manuscript fully available?

Reviewer #1: No

Reviewer #2: No

4. Is the manuscript presented in an intelligible fashion and written in standard English?

Reviewer #1: Yes

Reviewer #2: Yes

5. Review Comments to the Author

Reviewer #1: This paper innovates on the existing literature by integrating ecosystems benefits calculations into a hydro-economic optimization model, which is a useful contribution to the literature. It should also be noted that the authors show an in-depth knowledge of the literature and relevant methods. However, the paper contains too much general content on the literature and methods while not elaborating in enough detail the specifics of their own research. The introduction and methods sections contain too much general discussion of the literature and too many vague assertions. For example, paragraphs 2-4 in the introduction contain neither a) any citations nor b) any concrete examples. The paper would be improved by the judicious use of real-world examples/case studies to motivate the topic.

Relatedly, the Methods section should focus almost entirely on authors’ methodology rather than addressing the literature and the discussion of the Ebro basin. The discussion of the Ebro basin should be condensed and it belongs in its own section, which could include discussion of the data used. The Methods section could be dramatically reduced; I think that none, or almost none, of the content before the Environmental response function sub-section belongs in the Methods. Given the centrality of the WUA function to the paper, I think it merits a little more elaboration than given in lines 266-269. A diagram to aid reader intuition would be helpful.

A few sentences to explain the importance of the Ebro basin would be useful. While statistics on the basin are offered, there is little or no context offered as to the importance of the basin relative to Spanish agriculture or the EU as a whole. A map of Spain or perhaps the Iberian Peninsula with the Ebro basin called out would be useful if space allows.

The exposition of the mathematical programming model needs to be re-worked. There are a number of issues:

1. I’m not sure exactly what you mean by a “reduced-form” hydrological model in this context. Equations 1-3 give a system of equations that encapsulate the hydrology. To me a reduced-form model would collapse everything into a single equation. Not a major issue but worth considering if this is the best description.

2. The urban sector sub-model is problematic. How can you claim to maximize PS + CS given that prices are exogenous? That is, if you fix prices you know the equilibrium quantity. Moreover, as the optimization problem is stated, Q_s = Q_d is pre-determined. This all just collapses into a constraint requiring that urban demand must be met. The results further emphasize this point; urban water use and benefits are constant across all scenarios. I strongly suggest that the authors consider reducing equations 10-12 to a single constraint and/or restating exactly what the model is doing.

3. Sector sub-models are detailed as standalone models and then later combined into one integrated model. Are any of the models run in isolation, e.g., to calibrate one sector before being incorporated in the larger model? As currently written, the use of objective functions is confusing. That is, each sector has an objective function and then you refer to equations 1-15 in the larger model. But that model has its own over-arching objective function. So, I would probably change equation 4, for example, to read something like B = Sum(C X), removing the Max function.

4. The geography of the sectors and how they fit together is not at all clear. For example, how do cities (u) relate to ecosystem areas (s)? Figures 2-4 don’t seem to clarify this point. A network diagram would be very useful here.

5. How many observations were used to fit the regressions in Table 1? This may be indicated elsewhere in the text but the reader should not have to hunt them down. This info should be in the table or the caption.

6. I’m confused by the 3 levels of ecosystem values discussed in lines 438-447. The equations above this point explain how you calculate the environmental benefits. Given that this is endogenous to the model, why do you then bring up the 3 levels of values? It sounds like you’re overriding the values from the model. Can you explain this better?

7. Why do you not have a mathematical description of the institutional scenarios? That is, you should be able to modify the model by using an alternate version of the relevant equations or constraints. You may need to add additional equations/constraints that are scenario specific. By not clarifying in this manner, you leave it open to the reader’s interpretation how you implement the scenarios and whether it is a logical way of doing so.

8. The distinction between the baseline and the Institutional Cooperation scenarios is vague. The authors state in line 472 that “[t]his is the policy currently applied in the Ebro basin.” In other words, this is the same as the baseline scenario, right? The only difference being whether you have a drought or not. The way this is described and the subsequent results and discussion beg the question of whether you are conflating institutional and climate scenarios.

Currently, it is hard to evaluate the policy significance of your findings. The Results section could be significantly improved by clarifying and incorporating some of the issues raised above with regard to the Methods section. Specifically, I would suggest the following:

1. Why not elaborate the results for the alternate institutional scenarios under both Baseline weather and also under drought? Currently, it is impossible to completely separate the effects of institutional and climate factors. So, Table 2 could be split into two tables.

2. Given the relatively high dimension of scenarios and sectors to be detailed in the results, you might simplify a few aspects to facilitate interpretation:

a. Why not report results for alternate institutional arrangements relative to the appropriate baseline? That is, give the % change of the variables rather than their levels. This would help the reader understand the significance of institutions on the results.

b. Given that the urban sector is not actually optimized, I would remove it from the table and simply state the urban water use and benefit levels in the text or in the table caption. Then you clarify the trade-off between environmental and agricultural water use as presented in the table.

3. As far as I can tell, there is no motivation or context or the drought level chosen (40%). Please offer some rationale for that value.

4. The benefits for the Water Markets scenario appear to be strictly dominated by those from the Environmental Water Markets. Did/can you explain this and its significance?

The data visualization in Figures 2-4 could be improved. Figure 3, in particular, could be better. Why doesn’t this figure use bar charts showing stream flows by scenario for each location? Also, relative values (compared to baseline) would be better. Figures 2 and 4 are difficult to draw inference. Consider simplifying them and/or adding place names to the maps so that they may be related to Figure 3. Also, Figure 4 includes “Moderate” and “Severe” droughts. How are these defined? I see that on line 466 the level of 40% reduction is cited as a severe drought. Is that the level of drought considered in Table 2? There it is just mentions ”drought.” This is quite confusing.

Reviewer #2: This paper presents the development and application of a hydroeconomic model to understand interactions between environmental and human water uses under water scarcity and drought conditions in the Ebro basin, by explicitly accounting for environmental benefits linked to the river ecological status. This addresses an outstanding research gap in water resources systems modelling to support decision making. However, although the hydroeconomic modelling is rigorous there are concerns about quality/soundness of the information used to build and apply the model due to lack of detail and transparency about the methods and their limitations. Moreover, there are several aspects related to the written manuscript which should be addressed before the paper can be considered for publication. General comments on each section are provided below and more detailed comments are provided as annotations on the PDF file.

The introduction justifies the relevance and novelty of the research. However, the flow of ideas is confusing as it does not follow a clear structure: general issue > state of the art > research gap > aim and objectives. There is a general lack of references which are needed to support the research questions and selected approach.

The methods section describes how generally the assessment of environmental benefits is done supported by literature references more as an academic document to teach a subject than a scientific paper, which should not be part of the methods (perhaps summarised as part of the introduction and/or discussion). A methods section should be “recipe” that clearly and concisely describes how the results were obtained in this particular research so that other researchers could follow it to obtain the same results. This is somehow achieved from the sub-section “Ebro basin” onwards, although some details are missing which are mentioned in the PDF (e.g. ecosystem services considered and how they were valued – the study uses three valuation levels which are not appropriately justified; design and justification of the scenarios – the drought scenario is generated reducing 40% water inflows to the system but no justification is given about what are the basis for this). Relevant details (as they help demonstrate the validity/limitations of the modelling outputs) about how the optimisation was performed are missing. What algorithm was used to solve the objective function? Is the source code of the model available?

The results and discussion section (which should actually be two separate sections in the paper) does not include any kind of discussion about where the findings of this study sit in relation to previous studies in the field (there is just one citation in this section!) and the wider implications of the approach taken, including its limitations.

The conclusions are too long and are not actually a conclusion (i.e. the answer to the question) but a summary of the results. At the beginning of the study, the aim indicates what you set out to do (i.e. the economic and environmental impacts of droughts and water scarcity in the Ebro basin, and the social welfare that can be achieved under alternative water allocation policies considering environmental benefits), but the answer to that does not come through clearly enough to conclude the paper.

6. PLOS authors have the option to publish the peer review history of their article (what does this mean?). If published, this will include your full peer review and any attached files.

Reviewer #1: No

Reviewer #2: No

---

## [Author Response · Author response to Decision Letter 0]

12 Jan 2022

All reviewer and editor comments have been addressed in the Response Letter and the Cover Letter.

---

## [Decision Letter · Decision Letter 1]

11 Apr 2022

Integrating ecosystem benefits for sustainable water allocation in hydroeconomic modeling

PONE-D-21-31923R1

Dear Dr. Albiac,

We’re pleased to inform you that your manuscript has been judged scientifically suitable for publication and will be formally accepted for publication once it meets all outstanding technical requirements.

Kind regards,

Zaher Mundher Yaseen

Academic Editor

PLOS ONE

Additional Editor Comments (optional):

Reviewers' comments:

Reviewer's Responses to Questions

**Comments to the Author**

1. If the authors have adequately addressed your comments raised in a previous round of review and you feel that this manuscript is now acceptable for publication, you may indicate that here to bypass the “Comments to the Author” section, enter your conflict of interest statement in the “Confidential to Editor” section, and submit your "Accept" recommendation.

Reviewer #1: All comments have been addressed

2. Is the manuscript technically sound, and do the data support the conclusions?

Reviewer #1: Yes

3. Has the statistical analysis been performed appropriately and rigorously? 

Reviewer #1: Yes

4. Have the authors made all data underlying the findings in their manuscript fully available?

Reviewer #1: Yes

5. Is the manuscript presented in an intelligible fashion and written in standard English?

Reviewer #1: Yes

6. Review Comments to the Author

Reviewer #1: (No Response)

7. PLOS authors have the option to publish the peer review history of their article (what does this mean?). If published, this will include your full peer review and any attached files.

Reviewer #1: No

---

## [Editor Report · Acceptance letter]

18 Apr 2022

PONE-D-21-31923R1 

Integrating ecosystem benefits for sustainable water allocation in hydroeconomic modeling 

Dear Dr. Albiac:

I'm pleased to inform you that your manuscript has been deemed suitable for publication in PLOS ONE. Congratulations! Your manuscript is now with our production department. 

Kind regards, 

on behalf of

Dr. Zaher Mundher Yaseen 

Academic Editor

PLOS ONE